# Unilateral and Bilateral Post-Activation Performance Enhancement on Jump Performance and Agility

**DOI:** 10.3390/ijerph181910154

**Published:** 2021-09-27

**Authors:** Alejandro Escobar Hincapié, Carlos Alberto Agudelo Velásquez, Mariluz Ortiz Uribe, Camilo Andrés García Torres, Andrés Rojas Jaramillo

**Affiliations:** Instituto Universitario de Educación Física y Deportes (IUEFD), Universidad de Antioquia, Medellin 050034, Colombia; alejandro.escobarh@udea.edu.co (A.E.H.); mariluz.ortiz@udea.edu.co (M.O.U.); camilo.garciat@udea.edu.co (C.A.G.T.); andres.rojasj@udea.edu.co (A.R.J.)

**Keywords:** agility, single-leg exercise, double-leg exercise, warm-up, velocity-based resistance training

## Abstract

This study aimed to compare the effects of the post-activation performance enhancement (PAPE) of two different types of warm-ups, unilateral and bilateral, on the performance in vertical jumping and agility of healthy subjects with strength training experience. In the study, 17 subjects (12 men and 5 women) performed two different PAPE protocols: unilateral squat (UT) and bilateral squat (BT). The height of the subjects’ countermovement jump (CMJ) and the subjects’ time to perform the T-agility test (TAT) were measured before and after executing the PAPE warm-up. The squats were performed at a velocity of 0.59 m·s^−1^ with three sets of three repetitions, with a 3-min rest between sets and a 5-min rest after both uni- and bilateral PAPE warm-ups before taking the tests again. For statistical analysis, we applied ANOVA and calculated the effect size. The results showed that the PAPE for each case decreased the CMJ height but generated significant improvements in the total time taken for the T-agility test (*p* < 0.01); however, in both cases, the effect sizes were trivial. In conclusion, it is possible to observe that the PAPE, performed both unilaterally and bilaterally, negatively affects the performance in the vertical jump, showing moderate effect sizes. However, both PAPE protocols show performance benefits in agility tests, with a large effect size for the unilateral protocol and moderate for the bilateral protocol.

## 1. Introduction

Strength training plays a very important role in sports performance. Strength has been defined as the ability to produce voluntary muscle tension to achieve a goal [1]. In terms of strength measurement, the repetition maximum (RM) is a key tool when dosing training loads and measuring performance [2]. Although the dosage of loads starting at 1 RM is valid, the RM method has several drawbacks; these protocols can cause fatigue and insecurity [3]. Since RM percentages change daily due to different factors (inaccuracy in measuring the RM, fatigue, emotional state, etc.) [4], another method of dosing loads has emerged: the execution velocity or the effort indicator, which, based on the execution velocity, allows for the determination of training loads and the control of velocity loss, given that different velocity losses generate different adaptations [5]. Therefore, with a test of incremental loads mobilized at maximum velocity in the same training period, it is possible to determine training velocities and zones from linear regression in resistance training. This method of quantifying load has not been used when programming loads for post-activation potentiation, but it could be used due to its practicality.

In addition, before training or competition that requires strength, it is necessary to perform muscle activation, or, as it is otherwise known, a warm-up. Warm-ups can be defined as muscular actions performed before experiencing high muscular demands [6] in order to increase body temperature, range of motion, and speed of nerve transmission, as well as to reduce resistance to muscle viscosity [7]. The objective of a warm-up is to increase performance. However, the scientific term for the improvement of performance before competition or training is post-activation potentiation (PAP), which dates back to mid-1941 [8]. Post-activation potentiation is defined as an increase in muscle performance after a contraction, which could be the maximum voluntary contraction (MVC), a tetanic contraction, or a series of nerve impulses [9]. In previous research, some authors have suggested changing the name to PAPE (post-activation performance enhancement) since the phenomenon that occurs in PAP is very different from the analogous phenomenon in the sports field, and the time required to see the effects of each is not the same. This is because PAP peaks at around 28 s, while PAPE takes effect at 3 min [10] and its optimal results occur at 3–7 min [11]. Furthermore, the possible causes of PAPE differ from those of PAP. PAP improves performance through an improvement in the effectiveness of the contraction by a better coupling of actin and myosin, a phenomenon that has occurred in laboratory situations under electrostimulation. Meanwhile, PAPE occurs by increasing the temperature, the amount of water in the muscle fiber, and the number of activated motor units. This leads to increased motivation and a decrease in the perception of fatigue, and this phenomenon occurs with workload [10]. However, these two terms (PAP and PAPE) may not adequately describe all specific potentiation responses and mechanisms and can also be complementary in some cases [12].

There are different protocols aimed at PAPE for endurance and strength competitions. Thus, it is easy to find protocols to develop PAPE with an intensity of 90% of 1 RM [13]. However, there is strong evidence that PAPE is beneficial with loads ranging from 60 to 87% of 1 RM and 7–10 min rests after the generated stimulus [14].

In many sports, such as team sports, agility is a determining factor since performance in these sports depends on the decision-making that occurs while changes in direction are made and requires activation to achieve maximum speed [15].

Some studies have described the effects of using PAPE in warm-ups, including the improvement of movements that require great production of muscular power after the contractions in the conditions of almost maximum loads [9].

Likewise, positive results have been found when implementing unilateral training in sports activities with the aim of reducing muscular imbalances and improving strength levels [16]. Studies have also been conducted to compare the force generation produced unilaterally and bilaterally, resulting in greater energy production for unilateral exercises [16,17]. Nevertheless, improvements in performance have not yet been physiologically described or understood. One probable explanation is that there is a neural limitation during bilateral exercise that reduces the maximum force production, which is known as the bilateral deficit [14]. Ghahremani and Nazen’s research [17] on PAPE included three different warm-up groups: bilateral training (BT), unilateral training (UT), and a control group (CG). BT and UT involved performing a bilateral and a unilateral squat with a load of 90% of 1 RM, while the CG involved performing a traditional warm-up that included a 5-min run and 3 min of articular mobility. When measuring the jump height and maximum voluntary contraction isometrics, no significant differences between BT and UT in PAPE were found. Nonetheless, when comparing each method to a traditional warm-up, the differences were statistically significant. Various studies comparing BT and UT in the lower extremities have shown that UT is more beneficial for agility. However, when jumping, both types of training seemed effective as both UT and BT led to significant improvements after 6 weeks of training [18].

After considering all of this previous research, the aim of this study was to compare the effects of PAPE for two different types of warm-ups, unilateral and bilateral, on sports performance in vertical jumping and agility. Given this background, it was hypothesized that PAPE through unilateral exercise could be more beneficial for the T-agility test, while PAPE performed through bilateral exercise could be more beneficial for the vertical jump.

## 2. Materials and Methods

### 2.1. Experimental Approach

This study was designed to compare two types of warm-ups with high intensities by replacing the measurement of intensity with the execution velocity and not the percentage of 1 RM. This allowed us to observe the warm-ups’ effects on performance by measuring behavior in the vertical jump, which was evaluated through the CMJ, and on agility through the T-agility test (TAT).

A quantitative cross-sectional experimental study was conducted with the aim of determining the effects of PAPE—with the squat exercise performed unilaterally and bilaterally and programming the loads based on the execution velocity—on the performance of a countermovement jump (CMJ) and on agility, as measured by the TAT.

### 2.2. Participants

The study’s protocol adhered to the Declaration of Helsinki and was approved by the Institutional Review Board of University Catholica San Antonio of Murcia (Murcia, Spain), code CE012008. It was conducted on healthy individuals with strength training experience. Seventeen participants were selected (12 men and 5 women) with a mean age of 25 ± 1.6 years, a mean body mass of 70 ± 9.8 kg, a mean percentage of fat mass of 16 ± 2.1%, a mean height of 171 ± 7.5 cm, a mean strength and power training experience of 7.6 ± 2.3 years, and who performed minimum resistance training twice a week. The participants signed informed consent forms wherein the study’s protocol, benefits, and possible risks, as well as participants’ right to withdraw from the intervention whenever they wished, were explained. The inclusion criteria were (a) being between 18 and 30 years of age; (b) not having suffered from muscle or bone injury during the last year; and (c) having at least 2 years of strength training experience.

### 2.3. Procedure

In the first session, height was measured with a Seca stadiometer (Hamburg, Germany), body mass and fat percentage were measured with a Tanita BC 351 Scale (Tanita Corporation, Tokyo, Japan), and the load to be mobilized with the execution velocity was calculated. Before running the test with incremental loads, the weight to be moved was determined, and a standard 8-min warm-up was performed, which consisted of a 5-min run between 9 and 11 km/h and 1% inclination on the treadmill, followed by joint mobility and progressive runs. The initial load of the test was 50 kg for the bilateral squat and 25 kg for the unilateral squat. The weight was progressively increased by 10 and 5 kg, respectively, with 3-min rests after each set, until values close to or below 0.59 m-s^−1^ were obtained for the mean propulsive velocity (MPV) in the concentric phase; this speed value was chosen in order to determine the RM in a valid way [19]. The MPV differs from the mean velocity (MV) in the braking phase since the MPV eliminates this phase, and it is with the MPV that the percentages of the RM were associated [4]. Subsequently, the load to move was determined based on linear regression. The bilateral squat exercise was performed with a 90-degree flexion (corroborated by a cap for each person, which did not allow descending beyond these degrees, for both exercises), with the bar over the shoulders and facing forward. The eccentric phase was performed in a controlled manner, demanding maximum velocity in the concentric phase of the movement without taking the toes off the ground (Figure 1). The same protocol was used for the unilateral squat, but the leg was passively isolated and raised on a support that allowed the evaluated leg to perform 90-degree flexion, keeping this height for the whole movement (Figure 2). The entire protocol was performed on a Smith machine, which was chosen to improve the accuracy of the linear transducer (Chronojump, Barcelona, Spain) MPV measurement, restricting movement in the vertical direction. In this same session, participants were familiarized with the CMJ and the TAT.

In the second session, participants were randomized to begin the measurement in either the unilateral squat group (UT) or the bilateral squat group (BT). Then, the warm-up protocol described previously was implemented and the CMJ was executed. This type of jump was chosen because of its high correlation with performance [19]. For the measurement of CMJ performance, a contact platform was used (Chronojump, Barcelona, Spain). The protocol of this test was based on the fact that the athlete had to stand on the contact platform with his feet shoulder-width apart and his hands on the waist; then, at the signal of the evaluator, he had to descend at high speed, generating a knee flexion approximately at 90 degrees, and, at maximum speed, re-extend the ankle, knee, and hip to generate the jump at the maximum possible speed. The athlete had to keep his hands always on the waist, and, in the air, he had to stay in extension. For this test, 3 jumps were performed (with a 30-s rest between each one) and the best of these was taken.

With the TAT (Figure 3), agility was evaluated, partly because it is a valid tool to measure movements in different body planes and partly because of its reliability [20]. Electric timing gates (Kit Witty; Microgate, Bolzano, Italy) were used for its measurement. The test was performed 3 times and the best time was taken. Afterward, at the end of the 5-min rest period, the corresponding squat (BT or UT) was performed at a velocity of 0.59 m-s^−1^. As the estimated load was not mobilized at this velocity, the weight was modified. Then, 3 sets of 3 repetitions with a 3-min rest between sets were completed. Finally, at the end of the last set, a 5-min rest was completed. Subjects performed this cycle 3 times in each protocol, and the best results were recorded. This protocol was carried out by replicating the method applied in the study presented by Wilson, J.M. et al. [14].

In the last session, the counterbalance of the groups was performed, followed by the same warm-up protocol, and the proposed intervention was implemented.

### 2.4. Statistical Analysis

The data are presented as means ± standard deviations. The normality test used was the Shapiro–Wilk test (*p* ≥ 0.05). When normality was obtained in the data distribution, a parametric inferential statistics analysis was applied. For the comparison of the intragroup means (pre vs. post), ANOVA for paired means was used, considering differences with a value of *p* ≤ 0.05. In order to determine the effect of PAPE on performance, the effect size was categorized according to the following values: trivial (≤0.2), small (0.2–0.6), moderate (0.6–1.2), large (1.2–2.0), and very large (2.0). Intraclass correlation coefficients are presented with 95% confidence intervals. The statistical software used were SPSS (version 22.0; SPSS, Inc., Chicago, IL, USA) and JASP (JASP Team, Amsterdam, The Netherlands) software version 0.9.1.

## 3. Results

In Figure 4, it can be observed how both treatments produced similar effects in both the CMJ and the T-agility test, showing a decrease in performance in the jump and an increase in performance in the agility test.

Table 1 reinforces the results, showing a decrease in the CMJ performance groups in the unilateral and bilateral groups, 4.7% and 5.5%, respectively. Furthermore, this change in both groups is significant, with *p* values of <0.05 in both groups. The effect size was 0.72 (moderate) in the unilateral group and 0.84 (moderate) in the bilateral group.

On the other hand, the performance in the T-agility test increased, showing a statistical significance through the test (*p* < 0.05 in both groups), and the effect size was 1.3 (long) in the unilateral group and 0.99 (moderate) in the bilateral group.

## 4. Discussion

The main findings of this research were that PAPE yielded improvements in the agility test but reduced CMJ performance. These data are in contrast to the work of Ghahremani and Nazen [17], who observed a statistically significant decrease in CMJ height after PAPE. This is probably due to the difference in the protocols given that Ghahremani and Nazen conducted one set of two repetitions at 90% of the RM, whereas, in this study, the participants performed three sets of three repetitions and the intensity was quantified by means of the execution velocity (0.59 m-s^−1^, equivalent to 87% of 1 RM in a squat) according to the proposal of Badillo et al. [5]. Furthermore, in the intervention of Titton and Franchini [21], which aimed to assess the phenomenon of PAPE in young footballers, they used different percentages of 1 RM (between 40 and 100%) with rest combinations between 1 and 10 min and measured only the CMJ; in agreement with the present results, no statistically significant differences were found.

In a similar manner, our results are in line with a meta-analysis of PAPE on performance in vertical jumps with loads above 80% during squats, which was carried out by Dobbs et al. [11]. The results of that study showed negative values for the improvement of jumping performance after PAPE. In fact, this study suggested that rest is the variable with the greatest impact on these results since studies where the rest periods were less than 3 min and more than 7 min did not show changes in jumping performance or even reflected negative changes. Although our research included a 5-min rest, there were significant decreases in CMJ performance.

The results we obtained for the T-agility test are consistent with the improvements found in Fisher’s study [18], where two different BT and UT training periods were compared over 6 weeks. The improvements in agility were measured using the T-agility test, obtaining positive results for both groups with significant pre–post differences, with an effect size of 0.25 for BT and 1.48 for UT. In this research, where PAPE was evaluated, improvements were observed in both groups for the T-agility test. While, for Fisher [18], the difference between both groups was also significant, that was not the case in this research, probably because of the nature of the acute intervention.

Nevertheless, although the tests we used to measure agility are valid, recently, another tool, the agility deficit, has been proven to produce more sensitive results in the measurement of agility [22]. This method consists of finding the difference in the total time of the test minus the linear travel to obtain more accurate results regarding the time taken to perform the sprint, avoiding the interference of linear velocity. Though the agility deficit test was proposed using time, many authors use it with the difference in velocity [22,23]. Considering this, the changes obtained in the PAPE on the T-agility test in this study may have been affected by the participants’ improvements in linear velocity, as shown by Chatzopoulos et al. [24], who used loads of 90% of an RM with a 90-degree squat and obtained results showing significant improvements in velocity in 10- and 30-m sprints after PAPE.

Another possible explanation for the improvement in the T-agility test times is the effect of PAPE on decelerations. There was probably an improvement in decelerations in this test, but not necessarily in accelerations, which, in turn, would be connected to the CMJ given that there is fatigue in the muscles involved in the concentric phase of the jump. In addition to this, the performance of the muscles that intervene in an eccentric way in order to decelerate could have been improved by increases in blood flow and, subsequently, the amount of muscle water in response to intense exercise as there is a speculative increase in sensitivity to Ca^2+^ after intense exercise. This would cause a decrease in ionic strength (i.e., hypotonicity) within the muscle fibers, mainly as a result of the movement of water into the intracellular space during and after intense exercise [25]. We believe that the improvements in the T-agility test may also be owed to excitability at the spinal level. It has been found that, after brief and repetitive muscle contractions, there are increases in muscle activity [26]. Nuzzo et al. [26] observed that this increase in excitability was detectable immediately after the episode of contractions and slowly decreased to a basal state, but not before at least 20 min.

Moreover, it is also possible that increases in muscle temperature contribute to improving voluntary performance after PAPE. This was analyzed in studies that showed moderate improvements in performance (1–5%) [25,27,28].

Thus, changes in muscle temperature might theoretically explain a large percentage of the improvement in performance, as demonstrated by several studies showing increases of up to 5–10% [29,30,31]. Based on this evidence, the PAPE experienced after a conditioning activity (that was not preceded by an extended warm-up period) could be extensively, if not entirely, explained by an increase in muscle temperature, particularly when rapid rates of strength development or muscle shortening are critical for testing contraction performance.

In this study, the intensity per execution velocity was quantified, but the velocity loss or the index of effort represented for each individual in each situation was not considered since different losses of velocity produce different adaptations [32]. In addition, the rest interval may be another variable to consider in order to benefit from PAPE. This is why research considering different velocities and velocity losses that can cause different PAPE manifestations is necessary.

## 5. Conclusions

In conclusion, in this study, it is possible to observe that the PAPE performed both unilaterally and bilaterally negatively affects the performance in the vertical jump, showing moderate effect sizes. However, both PAPE protocols show performance benefits in agility tests, with a large effect size for the unilateral protocol and moderate for the bilateral protocol.

## 6. Practical Applications

Performing a squat with a mobilized load at 0.59 m-s^−1^ with a volume of three repetitions with 3-min rests between sets and a 5-min rest both unilaterally and bilaterally produces the same acute effects; however, these effects can be very important in sports where one of the main performance factors is agility.

## Figures and Tables

**Figure 1 ijerph-18-10154-f001:**
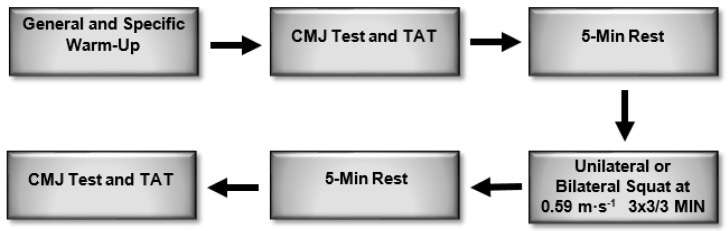
Protocol used for the intervention. CMJ: countermovement jump; TAT: T-agility test.

**Figure 2 ijerph-18-10154-f002:**
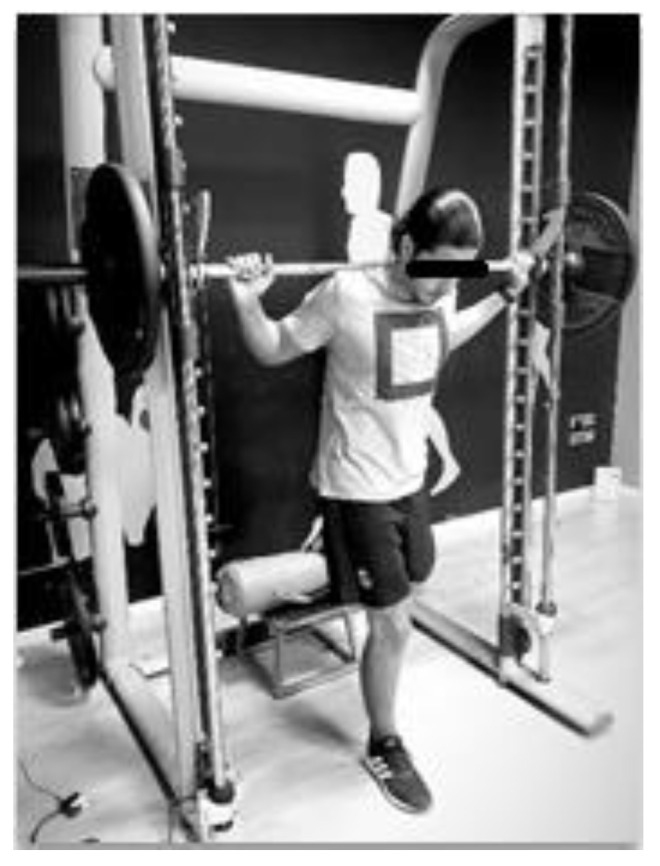
Unilateral squat (UT).

**Figure 3 ijerph-18-10154-f003:**
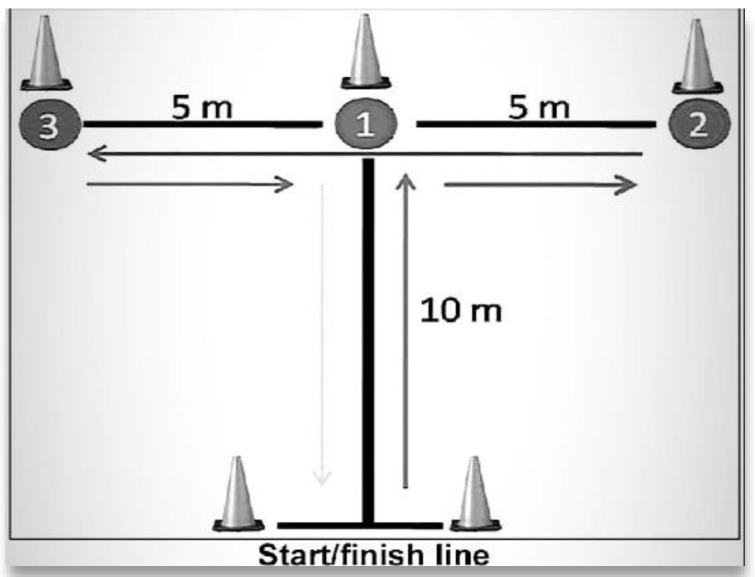
T-agility test.

**Figure 4 ijerph-18-10154-f004:**
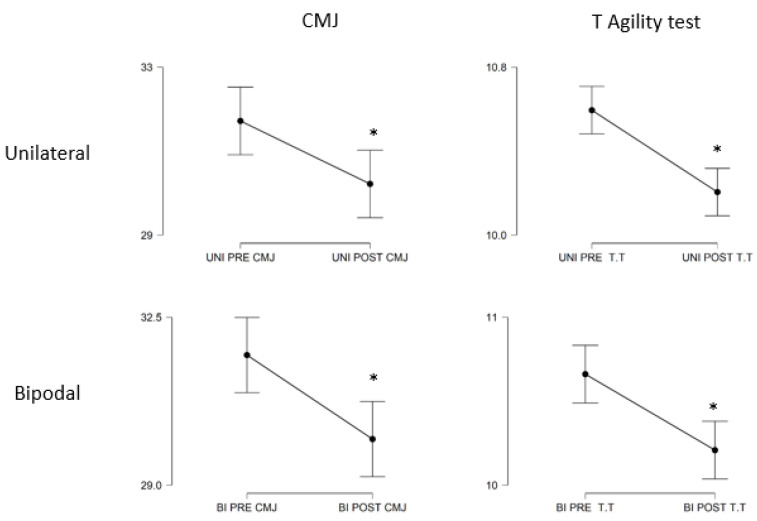
Comparison between the pre-test and the post-test of the PAPE groups in the counter-movement jump (CMJ) and in the T-agility test. * *p* < 0.05.

**Table 1 ijerph-18-10154-t001:** Comparison between the pre-test and the post-test of the PAPE groups in the countermovement jump (CMJ) and in the T-agility test.

**Unilateral Intervention (*n* = 16)**											
**Pre-Test**	**Post-Test**	**Δ%**	**F**	** *p* **	**η^2^**	**Mean Difference**	**95% CI for Mean Difference**		**SE**	**t**	**Cohen’s d**
**Mean ± SD**	**Mean ± SD**						**Lower**	**Upper**			
31.719 ± 6.518	30.219 ± 7.460	−4.7	7.881	0.013 *	0.344	1.500	0.361	2.639	0.534	2.807	0.702
10.595 ± 1.053	10.205 ± 0.907	−3.6	27.036	<0.001 *	0.643	0.390	0.230	0.550	0.075	5.200	1.300
**Bipodal Intervention (*n* = 16)**											
**Pre-Test**	**Post-Test**	**Δ%**	**F**	** *p* **	**η^2^**	**Mean Difference**	**95% CI for Mean Difference**		**SE**	**t**	**Cohen’s d**
**Mean ± SD**	**Mean ± SD**						**Lower**	**Upper**			
31.712 ± 7.351	29.961 ± 6.816	−5.5	11.339	0.004 *	0.431	1.752	0.643	2.861	0.520	3.367	0.842
10.661 ± 0.943	10.209 ± 0.817	−4.2	15.771	0.001 *	0.513	0.453	0.210	0.695	0.114	3.971	0.993

* *p* < 0.05

## Data Availability

https://docs.google.com/spreadsheets/d/1UKSVhVZs9A2SLzM3-pPcJBS6MYYOnXfz/edit?usp=sharing&ouid=102309446507164046296&rtpof=true&sd=true (accessed on 31 May 2021).

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
