# Peer review of "Unilateral and Bilateral Post-Activation Performance Enhancement on Jump Performance and Agility"

_ijerph, 2021, doi:10.3390/ijerph181910154_

Round 1

Reviewer 1 Report

General comments

The authors compared the effects of bilateral and unilateral squat exercises as PAPE protocols on vertical jump and agility performance measures. The results showed that the PAPE protocols impaired jump height whilst improved agility performance, although no differences were identified between conditions. This study has merit for practitioners and researchers alike, although some issues need to be addressed before it can be considered acceptable for publication.

Introduction

Lines 27-40: I’m struggling to follow the relevance of this paragraph to your study. This is all about RM testing, but how does this relate to your study topic?

Line 49: Avoid starting sentences with abbreviations

The introduction needs to justify the need to examining agility. At the moment, you only mention that agility has not yet been examined for BT and UT PAPE, but why is this important?

Materials and Methods

Line 95: What is TAT? It is abbreviated in the abstract, but this is the first time it is mentioned in the actual paper.

Line 117: Change warming to warm-up, and address this throughout the manuscript

Line 122: What was 0.59m/s used?

Lines 146-147: Why did you select 3 sets of 3 repetitions with 3-minute rest between sets for each PAPE protocol?

Line 148: Why did you select a 5-min rest period between the PAPE protocols and the performance protocols?

For statistical analysis, you should be conducting a two-way (condition x time) repeated measures analysis of variance.

Author Response

Medellin, 6 august 2021

Apreciated

Review 1

Cordial grettings.

Highly grateful for the contributions made to the article entitled: Unilateral and bilateral post-activation performance enhancement on jump performance and agility, the following points were improved according to your valuable insinuations:

Introduction

Lines 27-40: I’m struggling to follow the relevance of this paragraph to your study. This is all about RM testing, but how does this relate to your study topic?

I was initially:

Nowadays, strength training plays a very important role in sports performance. Strength has been defined as the ability to produce voluntary muscle tension to achieve a goal (Badillo & Ayestarán, 2002). In terms of strength measurement, the Repetition Maximum (RM) has been a key tool when dosing the training loads and measuring performance (Mayhew et al., 1995). Although the dosage of loads starting on 1RM is valid, the cons of using it are several, given that these protocols can cause fatigue and insecurity (Mayhew, Ware, & Prinster, 1993). Since, the RM percentages change daily due to different factors (inaccuracy in measuring the RM, fatigue, emotional state) (González-Badillo & Sánchez-Medina, 2010), a way of dosing the loads is through the execution velocity or the effort indicator, which, based on an execution velocity, allows to determine the training loads and control the velocity loss, given that different velocity losses generate different adaptations (Badillo, 2017). Therefore, with a test of incremental loads mobilized at maximum velocity in the same training it is possible to determine training velocities and zones from a linear regression.

Modified to:

Nowadays, strength training plays a very important role in sports performance. Strength has been defined as the ability to produce voluntary muscle tension to achieve a goal (Badillo & Ayestarán, 2002). In terms of strength measurement, the Repetition Maximum (RM) has been a key tool when dosing the training loads and measuring performance (Mayhew et al., 1995). Although the dosage of loads starting on 1RM is valid, the cons of using it are several, given that these protocols can cause fatigue and insecurity (Mayhew, Ware, & Prinster, 1993). Since, the RM percentages change daily due to different factors (inaccuracy in measuring the RM, fatigue, emotional state) (González-Badillo & Sánchez-Medina, 2010), a way of dosing the loads is through the execution velocity or the effort indicator, which, based on an execution velocity, allows to determine the training loads and control the velocity loss, given that different velocity losses generate different adaptations (Badillo, 2017). Therefore, with a test of incremental loads mobilized at maximum velocity in the same training it is possible to determine training velocities and zones from a linear regression in the resistance training. This way of quantifying the load has not been used when programming the loads when talking about post-activation potentiation but taking into account its practicality it could be used.

Line 49: Avoid starting sentences with abbreviations

I was initially:

(Ramsey & Street, 1941). PAP is defined as an increase in muscular performance after a contraction that could be the maximum voluntary contraction (MVC), a tetanic contraction, or a series 

Modified to:

1941 (Ramsey and Street, 1941). Post-activation potentiation is defined as an increase in muscle performance after a contraction that could be the maximum voluntary contraction (MVC), a tetanic contraction or a series

The introduction needs to justify the need to examining agility. At the moment, you only mention that agility has not yet been examined for BT and UT PAPE, but why is this important?

It was added:

In many sports such as team sports, agility is a determining ability, since it depends on the decision-making that occurs while changes of direction are made, situations that occur in all these sports and that also require activations to Maximum speed (Boyle, 2016).

Materials and Methods

Line 95: What is TAT? It is abbreviated in the abstract, but this is the first time it is mentioned in the actual paper.

I was initialy:

This allowed us to observe its effects on performance by measuring the behavior in the vertical jump which was evaluated through the CMJ and on the agility through the TAT.

Modified to:

This allowed us to observe its effects on performance by measuring the behavior in the vertical jump which was evaluated through the CMJ and on the agility through the t-agility test (TAT).

Line 117: Change warming to warm-up, and address this throughout the manuscript

I was initially:

Before running the test with incremental loads the weight to be move was determined, a standard 8-minute warming was performed which consisted of a 5-minute

Modified to:

Before running the test with incremental loads the weight to be move was determined, a standard 8-minute warm up was performed which consisted of a 5-minute

Line 122: What was 0.59m/s used?

I was initially:

with 3-minute rests after each set until values close to or below 0.59 m-s-1 were obtained in the mean propulsive velocity (MPV). The MPV differs from the mean velocity (MV) in the braking phase since the MPV eliminates this phase and it is with the MPV that the percentages of the RM are associated (González-Badillo & Sánchez-Medina, 2010)

Modified to:

with 3-minute rests after each set until values close to or below 0.59 m-s-1 were obtained in the mean propulsive velocity (MPV) in the concentric phase; this speed value is reached since with this it is possible to determine the RM in a valid way (Jiménez-Reyes et al., 2019). The MPV differs from the mean velocity (MV) in the braking phase since the MPV eliminates this phase and it is with the MPV that the percentages of the RM are associated (González-Badillo & Sánchez-Medina, 2010).

Lines 146-147: Why did you select 3 sets of 3 repetitions with 3-minute rest between sets for each PAPE protocol? Line 148: Why did you select a 5-min rest period between the PAPE protocols and the performance protocols?

I was initially:

As the estimated load was not being mobilized at this velocity, the weight was modified. Then, 3 sets of 3 repetitions with a 3-minute rest between sets were completed. Finally, at the end of the last set, a 5-minute rest was programmed to evaluate again the CMJ and the TAT.

Modified to:

As the estimated load was not being mobilized at this velocity, the weight was modified. Then, 3 sets of 3 repetitions with a 3-minute rest between sets were completed. Finally, at the end of the last set, a 5-minute rest was programmed to evaluate again the CMJ and the TAT, the subjects did it 3 times in each protocol and the best was taken, This protocol is carried out by replicating the study presented by Wilson, J. M et al (2007).

For statistical analysis, you should be conducting a two-way (condition x time) repeated measures analysis of variance.

I was initially:

The data are presented as mean ± standard deviation. The normality test used was Shapiro-Wilk (P≥0.05). When normality was obtained in the data distribution, a parametric inferential statistics analysis was applied. For the comparison of intra-group means (pre vs post) the Student’s t-test by paired means was used, considering differences with a value of p≤0.05. For the inter-group comparison (UT vs BT) the Student’s t-test for independent samples was used, considering differences with a value of p≤0.05. The statistical software used was the SPSS (version 22.0; SPSS, Inc., Chicago, IL, USA).

Modified to:

The data are presented as mean ± standard deviation. The normality test used was Shapiro-Wilk (P≥0.05). When normality was obtained in the data distribution, a parametric inferential statistics analysis was applied. For the comparison of intra-group means (pre vs post) the ANOVA for paired means was used, considering differences with a value of p≤0.05. In order to determine the effect of PAPE on performance, the effect size was used according to the following values: trivial (≤0.2), small (0.2-0.6), moderate (0.6-1.2), large (1.2-2.0), very large (2.0) (20). also presenting intraclass correlation coefficients with 95% confidence intervals The statistical software used was the SPSS (version 22.0; SPSS, Inc., Chicago, IL, USA) and JASP.

The clean text is also attached, as well as the one with all the corrections underlined.

Kind regards.

Reviewer 2 Report

Unilateral and bilateral post-activation performance enhancement on jump performance and agility

Dear Authors,

This is a very good and organized study using the PAPE phenomenon with unilateral and bilateral squat exercise on CMJ and T-Test agility performance test. I have to say that I am very excited when I see new studies like this. I comment the Authors for the effort and for the counterbalanced design. However, I am afraid that there are some major concerns that authors have to address before my second review.

Main concerns:

  1. Statistical analysis: Authors performed paired samples T-Test within groups and Student’s T-Test for independed samples (?) between groups. I strongly suggest changing the statistical analysis to Two-Way ANOVA (2 groups x 2 conditions) with repeated measures in order to ensure the significance level > 0.05. Then, perform paired samples T-Test between the percentage differences of CMJ and T-Test agility test.
  2. Emphasis on detail: There are many points where there is a clear lack of detail starting with references inside the text, units of measurement in the results etc.
  3. Limitations: This study was conducted on strength trained participants. There are several limitations that have to be mentioned inside the discussion. Also, authors did not measured any biological factor (i.e. muscle biopsies, muscle architecture, muscle stiffness, EMG etc.) which might have provided a better insight into the nature of the results. Having this in mind, and many more, I am sure that authors will present the limitations of their study.
  4. Practical Applications: Practical application needs more analysis. This is a strong aspect of the study. Ιn the way they are written they point to be more like conclusions. What does this study mean for coaches and strength/power athletes?

Abstract:

Abstract is well written. I suggest to authors to add the counterbalanced design of the study. This is a strong aspect of this study.

Line 14: Exponent -1.

Lines 14-15: Re-write how many sets and reps were performed for the unilateral procedure.

Line 17: No effect size is presented here. In addition, authors have to change the statistical analysis and add effect sizes as well.

Line 19: Use the abbreviation for the T-test agility test.

Keywords: Use comma between words.

Introduction:

Intro is good, but needs more flow and clarity.

Line 41: What capacity is that? Please, be specific.

Paragraph 2: There is a bad connection between lines 45 and 46. Beginning with warm up and then suddenly presenting the PAP history. Also, be more specific about the PAPE and the biological factors that may affect it (lines 57-58).

Paragraphs 3-4-5: Authors have to connect these paragraphs with better flow.

Line 75: Please unsure that all in text references are provided according to the journal’s guide of authors.

Line 87: What sports performance did the authors measured? Also, what is the hypothesis of the study?

General: There is a lack of the research questions inside the introduction. Research questions provide a good answer why this study was conducted and what we authors tried to search.

Material and Methods:

Methods are at a good level. I comment the authors for the counterbalanced design. Still, a figure of the design will be really helpful for readers.

Line 93: Change RM to 1-RM.

Line 95: TAT, Please clearly state the abbreviation before using it inside the text.

Participants:

What was the strength experience that participants had? Only strength training or power training as well? How many training sessions per week?

Line 105: Change to kg. Please apply this throughout the text.

Line 106: Change to cm. Please apply this throughout the text.

Line 112: Procedures: Please add intra-class correlation coefficients with 95% confident intervals and CV% for all measurements, including the squat exercise. Also, I suggest adding one more figure with the counterbalanced design and explain how the groups were separated.

Line 119: Was this applied for females as well?

Line 122: Is this speed includes only the concentric phase of movement? Also, correct the unit of measurement (m·s-1).

Line 126: How this angle was secured during the exercise? Was this angle the same for the unilateral exercise?

Line 138: How many trials were given to participants for the CMJ and TAT?

Lines 139-140: CMJ was chosen because of the link with fatigue or because the positive correlation between squats and vertical jump performance? Why fatigue?

Line 141: Be consistent with the abbreviations.

Lines 146-147: Please be more specific on how the unilateral training procedure was conducted.

Figure 1: Super.

Figure 2: Very good.

Figure 3: Super.

Statistical Analysis: Please follow my instructions above.

Results:

I think that results have to change according to the new statistical analysis. Please, provide F values and effect sizes. Also, add units of measurement in the numerical values (cm, sec, etc.).

Discussion:

According to the current results, discussion is in a good point. Authors have to perform the analysis and see again the discussion. Please, clearly present the limitations of the study and the practical applications.

Author Response

Medellin, 6 august 2021

Apreciated

Review 2

Cordial grettings.

Highly grateful for the contributions made to the article entitled: Unilateral and bilateral post-activation performance enhancement on jump performance and agility, the following points were improved according to your valuable insinuations:

  1. Statistical analysis: Authors performed paired samples T-Test within groups and Student’s T-Test for independed samples (?) between groups. I strongly suggest changing the statistical analysis to Two-Way ANOVA (2 groups x 2 conditions) with repeated measures in order to ensure the significance level > 0.05. Then, perform paired samples T-Test between the percentage differences of CMJ and T-Test agility test.

  1. Emphasis on detail: There are many points where there is a clear lack of detail starting with references inside the text, units of measurement in the results etc.

  1. Limitations: This study was conducted on strength trained participants. There are several limitations that have to be mentioned inside the discussion. Also, authors did not measured any biological factor (i.e. muscle biopsies, muscle architecture, muscle stiffness, EMG etc.) which might have provided a better insight into the nature of the results. Having this in mind, and many more, I am sure that authors will present the limitations of their study.

  1. Practical Applications: Practical application needs more analysis. This is a strong aspect of the study. Ιn the way they are written they point to be more like conclusions. What does this study mean for coaches and strength/power athletes?

With the totality of the corrections, it is expected that these concerns will have been addressed

Abstract:

Line 14: Exponent -1.

I was initially:

The squats were performed at a velocity of 0.59 m·s-1 with the volume of 3 sets of 3 repetitions

Modified to:

The squats were performed at a velocity of 0.59 m·s-1 with the volume of 3 sets of 3 repetitions

Lines 14-15: Re-write how many sets and reps were performed for the unilateral procedure.

Line 17: No effect size is presented here. In addition, authors have to change the statistical analysis and add effect sizes as well.

I was initially:

For the statistical analysis, we applied Student’s t-tests by paired and independent samples, and the effect size was calculated as well.

Modified to:

For the statistical analysis, we applied a ANOVA, and the effect size was calculated as well.

Line 19: Use the abbreviation for the T-test agility test. (as the conclusion was changed, it disappeared)

Keywords: Use comma between words.

I was initially:

Keywords: Agility. Single-leg exercise. Double-leg exercise. Warm-up. Velocity-based resistance training

Modified to:

Keywords: Agility, Single-leg exercise, Double-leg exercise, Warm-up, Velocity-based resistance training

Introduction:

Line 41: What capacity is that? Please, be specific.

I was initially:

Moreover, before a strength training or competition that requires this capacity, it is necessary to do a muscular activation, or, as it is commonly known, a warm-up.

Modified to:

In addition, before a strength training or competition that requires this ability, it is necessary to perform a muscle activation, or, as it is known, a warm-up.

Paragraph 2: There is a bad connection between lines 45 and 46. Beginning with warm up and then suddenly presenting the PAP history. Also, be more specific about the PAPE and the biological factors that may affect it (lines 57-58).

I was initially:

Although enhancing sports performance prior to the performance activity itself has been previously called post-activation potentiation (PAP), the origin of the term goes back to mid-1941 (Ramsey & Street, 1941).

Modified to:

It is clear that it seems that the objective of the warm-up is to increase performance, However, I term that is used scientifically to talk about the improvement of performance before competition or training is post-activation potentiation (PAP), the origin of the term dates back to mid-1941 (Ramsey and Street, 1941).

Paragraphs 3-4-5: Authors have to connect these paragraphs with better flow. (ingles)

Line 75: Please unsure that all in text references are provided according to the journal’s guide of authors. (no se por que esta corrección)

Line 87: What sports performance did the authors measured? Also, what is the hypothesis of the study?

General: There is a lack of the research questions inside the introduction. Research questions provide a good answer why this study was conducted and what we authors tried to search.

I was initially:

After considering all this previous research, the aim of this study was to compare the effects of PAPE of two different types of warming, unilateral against bilateral, on sports performance in vertical jump and agility.

Modified to:

Given this background, the authors wonder, is there a difference in the role of performing unilateral vs bilateral squat in sprint performance with change of direction and in jump height?

Material and Methods:

Line 93: Change RM to 1-RM.

I was initially:

of the RM. This allowed us to observe its effects on performance by measuring

Modified to:

of the 1-RM. This allowed us to observe its effects on performance by measuring

Line 95: TAT, Please clearly state the abbreviation before using it inside the text.

I was initially

This allowed us to observe its effects on performance by measuring the behavior in the vertical jump which was evaluated through the CMJ and on the agility through the TAT.

Modified to:

This allowed us to observe its effects on performance by measuring the behavior in the vertical jump which was evaluated through the CMJ and on the agility through the t-agility test (TAT).

Participants:

What was the strength experience that participants had? Only strength training or power training as well? How many training sessions per week?

Line 105: Change to kg. Please apply this throughout the text.

Line 106: Change to cm. Please apply this throughout the text.

I was initially

It was conducted on healthy individuals with strength training experience. Seventeen participants were selected (12 men and 5 women) with a mean age of 25 ± 1.6 years, a mean body mass of 70 ± 9.8 kilograms, a percentage of fat mass of 16 ± 2.1%, a mean height of 171 ± 7.5 centimeters, and a mean strength training experience of 7.6 ± 2.3 years. The participants signed the informed consent where the study’s protocol, benefits, possible risks, and right to withdraw from the intervention whenever they wished were explained. The inclusion criteria were: a) being between 18 and 30 years of age; b) not to have suffered from muscle or bone injury during the last year; and c) to have at least 2 years of strength training experience.

Modified to:

It was conducted on healthy individuals with strength training experience. Seventeen participants were selected (12 men and 5 women) with a mean age of 25 ± 1.6 years, a mean body mass of 70 ± 9.8 kg, a percentage of fat mass of 16 ± 2.1%, a mean height of 171 ± 7.5 cm, and a mean strength and power training experience of 7.6 ± 2.3 years; doing minimum resistance training twice a week. The participants signed the informed consent where the study’s protocol, benefits, possible risks, and right to withdraw from the intervention whenever they wished were explained. The inclusion criteria were: a) being between 18 and 30 years of age; b) not to have suffered from muscle or bone injury during the last year; and c) to have at least 2 years of strength training experience; Both men and women.

Line 112: Procedures: Please add intra-class correlation coefficients with 95% confident intervals and CV% for all measurements, including the squat exercise. Also, I suggest adding one more figure with the counterbalanced design and explain how the groups were separated. (falta la figura)

Line 122: Is this speed includes only the concentric phase of movement? Also, correct the unit of measurement (m·s-1).

I was initially

The weight was progressively increased by 10 and 5 kilograms respectively, with 3-minute rests after each set until values close to or below 0.59 m-s-1 were obtained in the mean propulsive velocity (MPV)

Modified to:

The weight was progressively increased by 10 and 5 kgs respectively, with 3-minute rests after each set until values close to or below 0.59 m-s-1 were obtained in the mean propulsive velocity (MPV) in the concentric phase

Line 126: How this angle was secured during the exercise? Was this angle the same for the unilateral exercise?

I was initially:

The bilateral squat exercise was performed with a 90-degree flexion, the bar over the shoulders, and facing forward.

Modified to:

The bilateral squat exercise was performed with a 90-degree flexion (corroborated by a cap for each person, which did not allow descending beyond these degrees, for both exercises), the bar over the shoulders, and facing forward.

Line 138: How many trials were given to participants for the CMJ and TAT?

I was added

Then, 3 sets of 3 repetitions with a 3-minute rest between sets were completed. Finally, at the end of the last set, a 5-minute rest was programmed to evaluate again the CMJ and the TAT, the subjects did it 3 times in each protocol and the best was taken,

Lines 139-140: CMJ was chosen because of the link with fatigue or because the positive correlation between squats and vertical jump performance? Why fatigue? Se borra la parte de fatiga

Line 141: Be consistent with the abbreviations.

I was initially:

With the T agility Test (Figure 3.) the agility was evaluated since it is a valid tool to measure movements in different body planes and due to its reliability (Raya et al., 2013).

Modified to:

With the TAT (Figure 3.) the agility was evaluated since it is a valid tool to measure movements in different body planes and due to its reliability (Raya et al., 2013)

Lines 146-147: Please be more specific on how the unilateral training procedure was conducted.

Results:

I think that results have to change according to the new statistical analysis. Please, provide F values and effect sizes. Also, add units of measurement in the numerical values (cm, sec, etc.).

The results section was modified

Discussion:

According to the current results, discussion is in a good point. Authors have to perform the analysis and see again the discussion. Please, clearly present the limitations of the study and the practical applications.

The discussion section was modified

The clean text is also attached, as well as the one with all the corrections underlined.

Kind regards.

Reviewer 3 Report

The PAP problme is still valid due to different results obtained by different authors. There are also no explanations that would be fully explained by the PAP mechanism.

Although the article may be interesting, it is written in a very colloquial way (no scientific language), e.g.: "The results obtained by us...", "races with changes of direction", "Before running the test with incremental loads the weight to be move was determined", etc. This makes the text very difficult to understand.

I suggest that the Authors first make a proper proof-reading correction and then send the work again. I would like to avoid a harmful assessment that could result from the poor linguistic level of the article.

Author Response

Medellin, 6 august 2021

Apreciated

Review 3

Cordial grettings.

Highly grateful for the contributions made to the article entitled: Unilateral and bilateral post-activation performance enhancement on jump performance and agility, the following points were improved according to your valuable insinuations:

A new correction was made to the language used in the text, and also authorizes in general the linguistic revision of the journal.

It is expected to facilitate its evaluation in this new versión

The clean text is also attached, as well as the one with all the corrections underlined.

Kind regards.

Round 2

Reviewer 1 Report

Thank you for addressing my comments. Unfortunately, the paper still requires substantial editing from a native-English speaker. I cannot recommend publication until this has been rectified.

Author Response

I hope that the native English editor assigned to me by the journal will follow your valuable suggestion, as we have already assumed the editing by an expert assigned by MDPI. I am enclosing the certificate of such fact.

Thank you very much

Reviewer 2 Report

Some minor comments.

  1. Authors added 12 tables from Anova analysis inside the results section. I suggest to keep only the important tables and the rest of the results to be mentioned inside the text. Also, a critical point here is that each table should be self-explanatory which does not applied for many of the tables inside the results. Please, clarify the labels of the tables. 
  2. Be  consistent with the abbreviations (i.e. line ).
  3. Practical Applications. Please, correct the title spelling. Also, this section is very important for the readers. Authors here must connect the theory with the practice and provide useful details to readers. This section should be much stronger but still is very weak.

Author Response

The proposed changes were made, and the text underlines where such changes were made. And attached.

1. The 12 tables were converted into 1 figure (Figure 4) lines 191-192 and 1 table (Table 1) lines 197-199 and were made more self-explanatory and their labels were clarified.

2. The consistency of abbreviations was revised.

3. The title was left as corrected in the revised English edition of MDPI, the practical applications were changed by pointing out the relevant data for readers, lines 301-303

Reviewer 3 Report

Thank you for correcting the text.

However, i feel that the paper still requires substantial editing from a native speaker. I don't understand why you use different words to describe the same things. Why are you not consistent? For example: unilateral and unipodal.

The distribution of Results with so many tables is now very difficult to read. Leave only important information.

The manuscript lacks a detailed description of the execution of the CMJ. Many performance factors affect the jump height (countermovement depth, arm swing), which may have an impact on the negative relationships you obtain. Complete the description of the procedures and add some information in the Discussion about the possible impact of performing the CMJ on the results obtained. Using this work should be helpful: https://link.springer.com/book/10.1007/978-3-030-31794-2

Author Response

The results were changed (underlined) page 5, we hope it was clearer and easier to read (in advance we highly appreciate having suggested such a change). As for the edition, we attach the certificate that it was made by the journal (MDPI native staff).

1. A new revision of the edition by MDPI has already been made and the issue has been corrected.

2. The 12 tables were converted into 1 figure (Figure 4) lines 191-192 and 1 table (Table 1) lines 197-199 and were made more self-explanatory and their labels were clarified.

3. It was not considered pertinent to expand the description of exercises that are fully identified with their name, and the relevant specific data and conclusions were added in lines 293-297.

Round 3

Reviewer 1 Report

Thank you for having the paper edited.

Lines 53-66: Please also consider the paper by Boullosa and colleagues (2020) which discusses the taxonomy of PAP in sports and training practices when you discuss the concept of PAPE. The paper can be found on PubMed (PMID: 32820135).

Lines 100-102: Please replace the last sentence of your Introduction with one that is hypothesis-driven. The sentence should start with, "It was hypothesized that..."

Line 296: The sentence starting with "however" should be upper case.

Author Response

Highly grateful for the contributions made to the article entitled: Unilateral and bilateral post-activation performance enhancement on jump performance and agility, the following points were improved according to your valuable insinuations:

  • Lines 53-66: Please also consider the paper by Boullosa and colleagues (2020) which discusses the taxonomy of PAP in sports and training practices when you discuss the concept of PAPE. The paper can be found on PubMed (PMID: 32820135).

R/ Included in line 68 and also in the bibliography as 5.

  • Lines 100-102: Please replace the last sentence of your Introduction with one that is hypothesis-driven. The sentence should start with, "It was hypothesized that..."

R/ The instruction was followed as shown in line 102.

  • Line 296: The sentence starting with "however" should be upper case.

R/ The instruction was followed as shown in line 307.

Reviewer 3 Report

It is not true that the very name CMJ explains fully how the jump was made. Finally, please complete the manuscript with the necessary information.

Author Response

Highly grateful for the contributions made to the article entitled: Unilateral and bilateral post-activation performance enhancement on jump performance and agility, the following points were improved according to your valuable insinuations:

  • It is not true that the very name CMJ explains fully how the jump was made. Finally, please complete the manuscript with the necessary information.

R/ The request in lines 162-169 is completed.  The protocol used was fully explained
